# DNS-BC: Fast, Reliable and Secure Domain Name System Caching System Based on a Consortium Blockchain

**DOI:** 10.3390/s23146366

**Published:** 2023-07-13

**Authors:** Tianfu Gao, Qingkuan Dong

**Affiliations:** State Key Laboratory of Integrated Service Networks, Xidian University, Xi’an 710071, China; tfgao@stu.xidian.edu.cn

**Keywords:** blockchain, consortium blockchain, DNS, DNS caching, DNS over KCP, DoK, DoT, DoH

## Abstract

The Domain Name System (DNS) is a fundamental component of the internet, responsible for resolving domain names into IP addresses. DNS servers are typically categorized into four types: recursive resolvers, root name servers, Top-Level Domain (TLD) name servers, and authoritative name servers. The latter three types of servers store actual records, while recursive resolvers do not store any real data and are only responsible for querying the other three types of servers and responding to clients. Recursive resolvers typically maintain a caching system to speed up response times, but these caching systems have the drawbacks of a low real-time performance, a poor accuracy, and many security and privacy issues. In this paper, we propose a caching system based on a consortium blockchain, namely DNS-BC, which uses the synchronization mechanism of the consortium blockchain to achieve a high real-time performance, uses the immutable mechanism of the consortium blockchain and our designed credibility management system to achieve up to a 100% accuracy, and has been combined with encrypted transmission protocols to solve common security and privacy issues. At the same time, this caching system can greatly reduce the traffic that name servers need to handle, thereby protecting them from Denial-of-Service (DoS) attacks. To further accelerate the data transmission speed, we have designed a new encrypted DNS protocol called DNS over KCP (DoK). The DoK protocol is based on the KCP protocol, which is a fast and reliable transmission protocol, and its latency can reach one-third of that of TCP when the network environment deteriorates. In our experiments, the transmission time of this protocol is about a quarter of that of the widely used encrypted protocols DNS over TLS (DoT) and DNS over HTTPS (DoH).

## 1. Introduction

A DNS is a fundamental component of the internet, as the majority of data transmission on the internet requires a DNS to resolve domain names into IP addresses. DNS servers are typically categorized into four types: recursive resolvers, root name servers, TLD name servers, and authoritative name servers. The typical resolution process involves a client sending a query request to a recursive resolver, which performs recursive resolution and returns the result to the client. During the recursive resolution process, the recursive resolver sends query requests to the root name servers, TLD name servers, and authoritative name servers, respectively, and eventually obtains a complete record. In the entire process, the recursive resolver acts like a middleman, helping users to quickly obtain resolution results, while the valuable data are stored in the name servers. To speed up response times, recursive resolvers typically maintain a caching system to store previously queried resolution results. When a client’s request hits a cache, the cached result is returned directly to the client, otherwise the resolver queries the name servers for the corresponding record. However, this caching system has many drawbacks [1], such as a low real-time performance, a poor accuracy, and many security [2,3,4,5] and privacy [6] issues.

To improve the performance of DNS caching systems and address security and privacy concerns, researchers have attempted to introduce blockchain [7] and encrypted transmission protocols into DNS caching systems. The research on introducing blockchain into DNS caching systems mainly focuses on how to increase the trustworthiness of caching through a trust sharing model [8], how to securely store DNS records using consortium blockchain [9], and how to efficiently allocate edge services based on geographical divisions to speed up access [10]. However, these systems, like traditional caching systems, still face issues such as non-real time caused by expired caches, a low accuracy due to untrusted data transmission channels and nodes, and the inability to protect valuable nodes in the system. The research on encrypted transmission protocols mainly focuses on how to resist attacks against DNS systems through encryption and authentication and how to address privacy concerns caused by unencrypted data transmissions. These protocols often use traditional symmetric encryption to encrypt transmitted data, but the process of negotiating keys and establishing transmission connections is too cumbersome, resulting in significant performance losses compared to traditional unencrypted DNS protocols [11].

To address these issues, we propose a caching system based on consortium blockchain [12], namely DNS-BC, which is maintained by all servers together in a ledger that stores cached records maintained by different nodes and the credibility score of each node maintained by authoritative nodes. The credibility score not only describes the trustworthiness of a node, but also indicates the probability of using the cache maintained by that node when a client’s request hits the cache. The calculation of the credibility score is described in Section 3.2. This caching system can ensure the accuracy and real-time performance by continuously verifying the cache, achieving up to 100% accuracy, which we will analyze and validate in Section 3.3 and Section 5.3, and eliminating the non-real time caused by expired caches. To further accelerate the data transmission speed, we have designed a new encrypted DNS protocol called DoK. The DoK protocol is based on the KCP protocol [13], which is a fast and reliable transmission protocol that can achieve a high transmission speed and a low latency. When the network environment deteriorates, the latency of KCP can reach one-third of that of TCP. In our experiments, the transmission time of DoK is about a quarter of that of the widely used encrypted DNS protocols DoT and DoH. The DoK protocol is designed to be compatible with the existing DNS system, and can be used to replace DoT and DoH. By introducing the DoK protocol into the caching system, based on the authentication feature of the DoK and the credibility management of the caching system, valuable nodes in the DNS system, namely name servers, can be protected from DoS attacks.

Overall, the major contributions of our work are summarized as follows:We propose a DNS caching system based on a consortium blockchain, which can archive up to 100% accuracy through the immutable mechanism of the consortium blockchain and our designed credibility management system and eliminate the non-real time caused by expired caches through the synchronization mechanism of the consortium blockchain.We propose a new encrypted DNS protocol called DoK, which is fast, reliable, and secure. In our experiments, the transmission time of this protocol is about a quarter of that of the widely used encrypted DNS protocols DoT and DoH. The DoK protocol is designed to be compatible with the existing DNS system and can replace DoT and DoH in traditional DNS systems.By introducing the DoK protocol into the caching system, valuable nodes in the DNS caching system, namely name servers, can be protected from DoS attacks.

This paper is organized as follows. Section 2 introduces the relevant research on how to apply blockchain technology in a DNS caching system and the defects of the existing caching system, as well as the current research status on encrypted DNS protocols and the performance bottleneck of these protocols. Section 3 presents the design of the DNS caching system. Section 4 describes the DoK protocol and its application in the caching system. Section 5 presents the experimental results and related analysis. Finally, Section 6 provides a conclusion of the results and proposes a direction for future research.

## 2. Related Work

### 2.1. Defects of the Existing DNS Caching System

In this section, we discuss the defects of existing DNS caching systems and summarize the efforts made by researchers.

The typical architecture of a DNS system is shown in Figure 1. In this system, a client sends a query request to a recursive resolver. Upon receiving the query request, the resolver first checks its cache. If the cache is hit, the resolver returns the corresponding record directly; otherwise, it performs an iterative query. The iterative query process consists of the resolver sequentially querying the root name server, TLD name server, and authoritative name server for the corresponding records and finally obtaining a complete record. For example, to resolve the A record of the domain name www.example.com (accessed on 6 July 2023), the resolver first queries the root name server for the address of the TLD name server that hosts the com domain, then queries the TLD name server for the address of the authoritative name server that hosts the example.com (accessed on 6 July 2023) domain, and finally queries the A record of www.example.com (accessed on 6 July 2023) from the corresponding authoritative name server. After the iterative query is completed, the resolver obtains a complete DNS record, returns it to the client, and updates the local cache as appropriate. The data transmission protocol is Do53, which runs on the top of UDP with no encryption and authentication.

This architecture has many problems, the biggest of which are the following:The transmission protocol Do53 is not authenticated and encrypted, resulting in many security [2,3,4,5] and privacy [6] issues.The accuracy of the cache is difficult to guarantee, and it is easy to be polluted [4].It is difficult to ensure the real-time performance of the cache. When authoritative name servers update the corresponding records, it cannot be guaranteed that the cache will be updated immediately.Valuable nodes in the network are not protected. Name servers store valuable data, making them the primary target of attackers. Attackers can use DNS flood attacks to attack name servers, causing them to be paralyzed.

Researchers have attempted to introduce blockchain technology into DNS caching systems to enhance their performance. However, the above-mentioned issues have not yet been resolved, and we will discuss them in Section 2.2. In addition, researchers have also proposed some encrypted DNS protocols to address the security and privacy issues which we will discuss in Section 2.3, but these protocols have an obvious performance bottleneck which we will discuss in Section 2.4.

### 2.2. Application of Blockchain Technology in DNS Caching Systems

In this section, we mainly focus on the research status of the application of the blockchain technology [7] in DNS caching systems, and point out the shortcomings of these studies, as well as our improvements.

Liu et al. [14] gave an overview of the application of blockchain technology in DNSs. Their study introduced the development and challenges faced by DNSs, including security, availability, and survivability issues, and then analyzed current DNS designs based on blockchain technology. There are several decentralized naming systems that have gained attention in recent years. Namecoin [15], which was launched in 2011, was the first cryptocurrency to use a blockchain for domain name registration. It is a peer-to-peer naming system that uses the same proof-of-work consensus mechanism as Bitcoin. Blockstack Naming Service (BNS) [16] is a naming system built using Blockstack’s separated control and data planes, which allows users to quickly bootstrap new nodes while having a high scalability and performance. Ethereum Name Service (ENS) [17] uses the Ethereum blockchain to store domain names and other digital assets. ENS domain names are human readable and can be used to access websites and services that are hosted on the Ethereum network. HandShake [18] uses a different consensus mechanism to Namecoin or ENS, and it also uses a different namespace. Handshake domain names are not subject to the same censorship and control as traditional DNS domain names. B-DNS [19] addresses two major issues in current blockchain-based DNS systems. It replaces the computation-heavy Proof-of-Work (PoW) protocol with a more efficient Proof-of-Stake (PoS) consensus protocol, and it solves the problem of inefficient queries by building an index of domains and designing a new index tree and search algorithm.

The DNS designs mentioned above all use blockchain to store and manage DNS records and other metadata, while this solves many of the problems in the current DNS, it also disrupts the existing architecture of DNSs, making it difficult to run on the existing infrastructure and thus difficult to promote. In particular, ENS-registered .eth domain names must be accessed in browsers that support the ENS. However, our DNS-based caching system does not disrupt the existing DNS architecture, but only improves upon it, making it more compatible with the existing infrastructure.

There are also some studies that have tried to introduce blockchain technology into DNS caching. Zhong et al. [8] proposed a DNS Cache Resources Trusted Sharing Model that utilizes a consortium blockchain to improve the credibility of DNS resolution results. They introduced the consortium blockchain as the carrier of the peer-to-peer network to reduce the impact of illegal access and complicity tampering on the DNS cache credibility. They also proposed a method for calculating the credibility score of DNS resolution nodes and a trust-based incentive mechanism to reduce the impact of free-riding behavior. Chen et al. [9] presented DagGridLedger, a sharded DAG blockchain that provides a scalable big data architecture for trustworthy DNS caching management. They proposed a radical new architecture that combines blockchain sharding and DAG techniques to achieve a high performance and scalability. They introduced DagGrid, a DAG-based blockchain consensus algorithm that improves the overall throughput of DNS queries. Choncholas et al. [10] proposed GeoENS, a prototype based on the Ethereum blockchain that provides accurate fine-grained geographic localization of edge services and fast look-ups. They identified the limitations of traditional DNS caching systems in meeting the requirements of edge computing and proposed blockchain-based solutions. They introduced GeoENS, which features a novel record organization for smart contracts, push-based record invalidation, and a look-through cache.

However, these studies mainly focus on improving the end-user experience, without considering the protection of valuable nodes in the DNS caching system or addressing the issues of low accuracies and insufficient real-time performances in traditional DNS caching systems mentioned in Section 2.1. Our newly designed consortium-blockchain-based DNS caching system can effectively protect name servers from DoS attacks while ensuring both security and reliability, significantly improving the accuracy and real-time performance of the caching system.

### 2.3. Current Research Status on Encrypted DNS Protocols

In this section, we focus on the research status of encrypted DNS protocols, point out the shortcomings of existing solutions, and propose our improvements.

The traditional DNS protocol, namely Do53 protocol, is not authenticated and encrypted, resulting in many security and privacy issues. To address the security issues caused by the lack of authentication, IETF engineers added a security extension to the Do53 protocol, called the Domain Name System Security Extension (DNSSEC) [20]. The DNSSEC uses digital signatures based on public key encryption to enhance the DNS verification strength, including data source verification and data integrity protection.

However, the DNSSEC only prevents tampering of transmitted data, and the entire transmission link is still not encrypted, meaning that attackers can eavesdrop transmitted data, which poses significant privacy issues. To encrypt the transmission link, researchers proposed the DoT protocol [21], which uses TCP + TLS to encrypt the transmission link and prevent attackers from eavesdropping on transmitted data. This protocol uses port 853.

While DoT implements authentication and encryption, the use of the uncommon port 853 makes its traffic too distinctive, making it easy to be detected and intercepted. To disguise the traffic and prevent attackers from intercepting transmission traffic, researchers proposed the DoH protocol [22], which adds an HTTP layer on top of TCP + TLS and uses port 443, making DoH traffic appear like normal HTTPS traffic and effectively avoiding detection by attackers.

To speed up transmission, researchers later proposed the DNS over QUIC (DoQ) [23] protocol, which uses the Quick UDP Internet Connection (QUIC) protocol to transmit DNS messages. The QUIC protocol is a transport layer protocol developed by Google. It is a modern, secure, and multiplexed protocol that aims to improve the performance of HTTP/2 and TLS over unreliable networks. Overall, the QUIC protocol is designed to provide a better performance, reliability, and security than TCP.

The above-mentioned encrypted DNS protocols all use different underlying protocols to transmit DNS messages, which are mature and stable existing protocols, but they all have certain performance issues [11]. Zhang et al. [24] proposed a UDP-based encrypted DNS protocol: the DNS Data Encryption Algorithm (DNSDEA). This protocol integrates the PKI encryption system with the DNS protocol and significantly reduces the DNS query latency compared to existing DNS encryption protocols. However, the downside of this protocol is that its underlying protocol uses UDP, which cannot guarantee transmission stability and reliability.

To address performance issues while ensuring security and privacy, we propose a fast, stable, and reliable encrypted DNS protocol, DoK, which can achieve secure, stable, and reliable data transmission at four times the speed of DoT and DoH.

### 2.4. Performance Bottleneck of DoT and DoH

In this section, we analyze the performance bottleneck of the widely used encrypted DNS protocols DoT and DoH.

Hounsel’s research [11] shows that compared to the unencrypted Do53 protocol, the response time of DoT and DoH is usually more than 4 times slower. The main reason for such a huge performance gap is that the handshake process of DoT and DoH is too cumbersome, resulting in a lot of wasted round trip time (RTT). In telecommunications, the round trip delay or round trip time is the amount of time it takes for a signal to be sent plus the amount of time it takes for acknowledgement of that signal having been received.

Taking DoT as an example, let us analyze how long a complete DNS resolution takes under ideal conditions without packet loss. The response time of the DoT protocol is divided into four parts: TCP handshake, TLS initialization, TCP transmission time, and query time. TCP requires a three-way handshake to establish a connection between the client and server before sending the data, which will take $1.5t_{RTT}$, where $t_{RTT}$ is the average value of RTT. Similarly, to terminate or stop the data transmission, a four-way handshake is required, which will take $2t_{RTT}$. TLS 1.2 initialization requires $2t_{RTT}$, while TLS 1.3 initialization requires $1t_{RTT}$. Let us assume that the faster TLS 1.3 is used, which requires $1t_{RTT}$. During data transmission, the client first sends data to the server, and the server returns a response and an ACK. After receiving data from the server, the client also responds with an ACK. This process takes $1.5t_{RTT}$. Assuming that the query request hits the cache, the resolver queries data from its memory, which takes $t_{query}$ time. Therefore, the total response time required for the DoT protocol is:(1)$$t_{DoT}=6t_{RTT}+t_{query}$$

For the Do53 protocol, the query process is relatively simple. The Do53 protocol mostly runs on top of UDP. In the case of no packet loss, a query only requires three steps: the client sends a request to the server, the server queries the result, and the server returns the response to the client. The total time required is:(2)$$t_{Do53}=t_{RTT}+t_{query}$$

We can see that the performance bottleneck of DoT is the cumbersome handshake process, and the performance bottleneck of DoH is the same as that of DoT. To solve this performance bottleneck, we propose the DoK protocol. In Section 4, we will describe the design of this protocol in detail and analyze how it overcomes this bottleneck.

## 3. DNS Caching Systems Based on Consortium Blockchains

### 3.1. Architecture of the Caching System

In this section, we mainly focus on the overall design, workflow, and corresponding algorithms of our designed DNS caching system DNS-BC.

The architecture of DNS-BC is presented in Figure 2. Compared to the typical architecture of a DNS system shown in Figure 1, the data transmission protocol is our designed DoK protocol, and the cached records are stored on the consortium blockchain, which is maintained by all legitimate members, including authenticated name servers and resolvers. Each member of the consortium blockchain has its own credibility score *c*, ranging from 0 to 100%. The credibility score of recursive resolvers is less than 100%, while the credibility score of name servers is always 100%, as they are the owners of the data. This credibility score not only describes the trustworthiness of a node, but also indicates the probability of using the cache maintained by that node when a client’s request hits the cache. There is a probability of *c* to use the cache, and a probability of 1−c to continue with iterative queries. This means that for cached data maintained by recursive resolvers, there is always a certain probability of not using them, while for cached data maintained by name servers, they will always be used. The workflow of the resolver is shown in Figure 3.

When the resolver receives a request from the client, it first checks whether the cache on the consortium blockchain is hit. If the cache on the consortium blockchain is not hit, it means that no node is maintaining that record yet. At this time, an iterative query is performed, and a new cache is added as appropriate, and then the query result is returned to the client. Whether to maintain this new cache is entirely up to the consortium blockchain node itself. For resolver nodes, they can choose to maintain more caches to improve their credibility score or choose to maintain fewer caches to reduce server pressure. For name server nodes, they can choose to maintain more caches to reduce the number of requests to be processed or choose to maintain fewer nodes to better protect user privacy and valuable business data. If the query request from the client hits the cache, there is a probability of *c* to use the cache, and a probability of 1−c not to use the cache. If the cache is used, the corresponding record is returned directly to the client. Otherwise, an iterative query is performed, and the query result is compared with the cache on the chain before being returned to the client. If the query result is consistent with the cache result on the chain, no action is taken. Otherwise, this issue is reported to the authoritative nodes, which will determine whether the cache is incorrect and whether the credibility of the node maintaining the cache should be reduced. Name servers only receive and process traffic from authenticated consortium blockchain members, and any other traffic will be directly discarded. They are also obliged to update the corresponding cache on the chain when a record for a domain name is updated by the administrator.

The overall function modules of the caching system are shown in Figure 4.

The data storage module refers to the data stored on the consortium blockchain, which consists of three parts: DNS records, the credibility of each node, and the information of each member, especially the authoritative nodes responsible for maintaining credibility and conducting membership authentication.

The data handling module consists of a records querier, a cache updater, and a cache validator. The records querier is responsible for querying data on the consortium blockchain, the cache updater decides whether to update the cache and updates it, and the cache validator is responsible for verifying the correctness of the cache on the consortium blockchain and reporting it to the authoritative nodes.

The service module consists of a credibility manager, membership authentication, and certificates manager. The credibility manager is responsible for maintaining the credibility of each node, membership authentication decides whether to allow new members to join, and the certificates manager is responsible for issuing certificates, which will be used in the DoK protocol.

### 3.2. Maintenance of the Consortium Blockchain

In this section, we will introduce the maintenance of the consortium blockchain, including credibility management, membership authentication, and certificate management.

Nodes in the consortium blockchain are divided into two types: authoritative nodes and ordinary member nodes. Authoritative nodes are elected periodically and are responsible for maintaining their own DNS cache, as well as managing new member admissions, credibility management, membership authentication, and certificate management. Ordinary member nodes only maintain the DNS cache. Authoritative nodes differ from conventional consortium blockchain administrators such as the administrator user in Hyperledger Fabric [25]. Authoritative nodes are multiple nodes rather than a single user, and they are not predetermined like administrator users. They maintain a series of ledgers to store management-related data, and the administrator user is responsible for periodically executing the corresponding operations based on this data, such as member admissions. The process of electing authoritative nodes is very simple. Member nodes who wish to become authoritative nodes can sign up for the election, and the top *N* nodes with the highest credibility score will be elected as authoritative nodes. The workflow of electing authoritative nodes is shown in Figure 5, where $c_{i}$ indicates the credibility score of node *i*, and $c_{N}$ indicates the Nth highest score among all participating nodes in the election.

Authoritative nodes will be responsible for credibility management, which involves managing the credibility score *c* of each member. Each member must try to improve their credibility score as much as possible, because when $c<c_{min}$, the authoritative nodes will kick that node out of the consortium blockchain.

Assuming the set of resolver nodes in a consortium blockchain is $\Lambda_{r}$ and the set of name servers nodes is $\Lambda_{n}$, for node *i*, its credibility score $c_{i}$ is defined as follows:(3)$$c_{i}=\left\{\begin{array}{cc} 2^{\frac{-q}{1-q}}-1 & i\in \Lambda_{r}\\ 1 & i\in \Lambda_{n} \end{array}\right.$$

When $i\in \Lambda_{n}$, the node is a name server and its $c_{i}$ is always 1 because it is the owner of the DNS records. When $i\in \Lambda_{r}$, the node is a recursive resolver and $0\leq c_{i}<1$, where *q* is a number in the range of [0,+∞). As *q* approaches 0, $c_{i}$ approaches 0, and as *q* approaches +∞, $c_{i}$ approaches 1. The formula for calculating *q* is as follows: (4)$$q=\zeta \frac{t_{c}K}{Me^{\sum {}_{j=1}^{K}t_{j}}}$$
where ζ is a constant greater than 0, which controls the growth rate of credibility. The larger ζ is, the faster the credibility grows. $t_{c}$ (s) represents the duration of time during which no errors have occurred, i.e., the time elapsed from the last error report to the present. *K* represents the number of caches maintained by the current node, while *M* represents the total number of cache accesses. $t_{j}$ represents the time elapsed since the *j*-th cache maintained by the current node has expired. If the cache has not expired, $t_{j}$ is equal to 0.

We put the variables that positively incentivize credibility in the numerator and the variables that negatively incentivize credibility in the denominator. For the following variables:$t_{c}$ positively incentivizes credibility because when a node’s cache is error free for a long time, it indicates that the node is reliable.*K* positively incentivizes credibility because when a node is willing to maintain more cache, it means that the node is willing to take on more responsibility, and its credibility should be higher.*M* negatively incentivizes credibility because when a node maintains only a small amount of cache but accesses a large amount of cache maintained by other nodes, it indicates that the node is not willing to take on more responsibility but wants to use more resources, and its credibility should be lower.$t_{j}$ negatively incentivizes credibility because when a node’s cache expires, it indicates that the node is unreliable.

The authoritative nodes are also responsible for membership authentication and certificate management. Membership authentication refers to the admission and removal of members. If a member’s credibility score *c* is less than the minimum credibility score $c_{min}$, they will be removed. If a node wants to join the consortium blockchain, its legal identity needs to be verified. This is done by directly sending some DNS requests to query the corresponding DNS records. If these records are valid, the node is considered legitimate. Certificates are used in the DoK protocol, and authoritative nodes are responsible for issuing certificates to verify the identity of nodes in the consortium blockchain.

### 3.3. Security and Performance Analysis of the Caching System

In this section, we analyze the security and performance of DNS-BC and answer how it solves the three problems mentioned in Section 2.1.

The accuracy of traditional DNS caching systems is low because they are easily polluted. In our designed caching system, encrypted and authenticated data transmission protocols are adopted to prevent attacks on data transmission, and a consortium blockchain is used to store data, ensuring that data cannot be tampered with. At the same time, in the credibility management mechanism, the credibility score of the recursive resolver is always less than 1 which is guaranteed by Formula (Equation 3), so the correctness of the caches maintained by the recursive resolver will be continuously verified by other nodes, ensuring that these caches always have a high credibility.

When all caches are maintained by name servers, the credibility score of all caches is 100%, which is guaranteed by Formula (Equation 3). At this time, the cache hit rate is also 100%, because as described in Section 3, the credibility score not only represents the credibility of a node, but also represents the probability of using the cache maintained by that node when a query request hits the cache. We will verify in Section 5.3 that DNS-BC can indeed achieve a cache hit rate of 100%. Furthermore, since name servers themselves store DNS records, the cache data from them can be considered completely correct. In addition, the consortium blockchain ensures that data cannot be illegally tampered with, so in this case, when all caches are maintained by name servers, the accuracy of the cache can be considered as 100%.

Traditional DNS caching systems also have the issue of low real time caused by expired caches, which can also be well solved in DNS-BC. Since the correctness of the caches in DNS-BC will be continuously verified by other nodes, expired caches can be detected in time. In addition, under ideal conditions, if all caches are maintained by name servers, since name servers are obliged to update caches in a timely manner when DNS records are changed by administrators, the low real time problem caused by expired caches can be completely solved.

DNS-BC can also be combined with the DoK protocol to protect valuable nodes in the system, namely name servers. Since recursive resolvers do not store data and only act as middlemen to help users obtain resolution results faster and recursive resolvers are usually maintained by Content Delivery Network (CDN) providers or cloud service providers such as Cloudflare and Google, they have a certain ability to resist DoS attacks. In contrast, name servers need to be protected more from DoS attacks. In DNS-BC, name servers only accept traffic from authenticated consortium blockchain nodes, and all other traffic is directly discarded, dramatically reducing the amount of traffic that needs to be processed. Since the identities of all members in the consortium blockchain are known, when a malicious node launches a DoS attack against a name server, it can be directly kicked out of the consortium blockchain. In addition, since the credibility score of the name server is 100%, the caches maintained by the name server will be directly used by the recursive resolver, and the recursive resolver will never send requests to the name server to query the record again. Therefore, the more caches maintained by the name server, the less traffic it needs to process. When all records owned by a name server are put on the consortium blockchain, this name server does not need to process any requests. We will verify this in the experimental Section 5.

## 4. The DoK Protocol

### 4.1. The Format of the DoK Message

This section provides a detailed description of the format of the DoK message.

The message format of DoK is shown in Figure 6, where (a) illustrates the DoK message format and (b) illustrates the format of the flags field.

In Figure 6a, the detailed explanations for each field are as follows:Transaction ID (16 bits): This field is used to match the response with the request sent from the client side. Matching is carried out by this field as the server copies the 16-bit value of identification in the response message so the client device can match the queries with the corresponding response received from the server side.Flags (16 bits): This field is used to identify certain attributes of the current message, and its specific structure is shown in Figure 6b, which we will discuss shortly.Questions (16 bits): This field specifies the number of questions in the Queries section of the message.Answers RRs (16 bits): This field specifies the count of answer records in the Answers section of the message. This section has a value of 0 in query messages, and is available only in response messages.Authority RRs (16 bits): This field gives the count of the resource records in the Authoritative Name Servers section of the message. It gives information that comprises domain names about one or more authoritative servers. This section has a value of 0 in query messages, and is available only in response messages.Additional RRs (16 bits): This field holds the number of Additional Records, which are used to keep additional information to help the resolver. This section has a value of 0 in query messages, and is available only in response messages.Queries (variable length): This field contains the queries made by the client. The number of queries is specified in the Questions field.Answers (variable length): This field contains the server’s answers to the queries made by the client, which is available only in response messages. The number of answers is specified in the Answers RRs field.Authoritative Name Servers (variable length): This field contains the domain names of the authoritative servers that are responsible for managing a specific domain name. The number of authoritative servers is specified in the Authority RRs field.Additional Records (variable length): This field contains additional information that can be used to help the resolver. The number of additional records is specified in the Additional RRs field.Client Public Key (variable length): This field contains the public key of the client, which is available only in query messages.

In Figure 6b, the detailed explanations for each field are as follows:QR (Query/Response, 1 bit): This subfield is used to indicate whether the message is a query or a response. A value of 0 indicates a query message, while a value of 1 indicates a response message.Opcode (2 bits): This subfield defines the type of query carried by the message. This value is repeated in the response. It can have three possible values: 0 for a standard query, 1 for an inverse query that finds the domain name from the IP address, and 2 for a server status request.AA (Authoritative Answer, 1 bit): When set to 1, it indicates that the name server is authoritative. It is only used in response messages.VF (Verify, 1 bit): This subfield is usually 0. When set to 1, it indicates that the server has been subjected to a DoS attack and requires source authentication from the client. The source authentication process is simple; when the client receives the request, it initiates a three-way handshake to establish a TCP connection. If the connection is successful, the source authentication is passed; otherwise, it fails. For zombie networks that like to send brute force packets, source authentication requests can easily be ignored, resulting in their IP addresses being blacklisted.RD (Recursion Desired, 1 bit): When set to 1, it indicates that the server is expected to perform a recursive query. When set to 0, it indicates that the server is expected to perform an iterative query.RA (Recursion Available, 1 bit): This subfield indicates whether recursive queries are available. It only appears in response messages, and a value of 1 indicates that the server supports recursive queries.CID (Certificate Identity, 4 bits): This subfield is used to identify the ID of the current temporary certificate. The DoK protocol requires the server to issue a temporary certificate to establish a connection. We will discuss the workflow and the usage of temporary certificates in Section 4.2.UC (Update Certificate, 1 bit): When set to 1, it indicates that the server requires the client to update the temporary certificate. This subfield is always 0 in query messages.Rcode (4 bits): This subfield is used to denote whether the query was answered successfully or not. There are several values of rcode, each with their error status.–A value of 0 indicates no error.–A value of 1 indicates that there is a problem with the format specification.–A value of 2 indicates a server failure.–A value of 3 refers to a name error, which implies that the name given by the query does not exist in the domain.–A value of 4 indicates that the request type is not supported by the server.–A value of 5 refers to the non-execution of queries by the server due to policy reasons.

### 4.2. The Workflow of the DoK Protocol

In this section, we will introduce the workflow of the DoK protocol.

The DoK protocol is based on the KCP protocol [13], which is a fast and reliable transport protocol designed specifically for online games, live streaming, and other real-time communication applications. It is built on top of the UDP and provides features such as packet-level error correction, congestion control, and fast retransmission. KCP can effectively reduce the latency and packet loss while maintaining a high level of reliability, making it an ideal choice for applications that require real-time communication.

KCP does not require a handshake to establish a connection. It is designed to negotiate initialization parameters through the upper-layer protocol. To establish a KCP connection, at least three parameters are required: the IP address, port number, and conv number. The conv number is used to identify a session, while other parameters such as RTT and RTO (Retransmission Time Out) have default values that are dynamically updated during data transmission. In DNS resolution, the server’s IP address and port number are known, so the client can assume that the server has initialized a connection and can send data directly. The conv number is randomly generated, and it only needs to ensure that no connection has been established with the same conv number in the current client. If the server receives a KCP packet and finds no corresponding connection, it initializes a connection based on the sending IP address, port number, and conv number in the KCP message and responds with an ACK (ACKnowledgment). Otherwise, the KCP retransmission mechanism is triggered. If the client does not receive an ACK message from the server, it assumes that the server has completed initialization and triggers the retransmission mechanism, continuously sending packets until the server responds with an ACK.

Before establishing a DoK connection, both the client and server need to generate a pair of public and private keys for data encryption and decryption. The server’s public key needs to be issued to the client, which requires authenticating the server’s identity through a digital certificate. In the DoK protocol’s design, this digital certificate is called a temporary certificate, which is only used for communication between the client and server. The temporary certificate is authenticated by a pseudo-root certificate issued by a trusted management agency and authenticated by the client’s built-in CA (Certificate Authority) root certificate. In DNS-BC, the certificate issuance process consists of the authoritative node first applying for a pseudo-root certificate from the CA and then issuing temporary certificates to various nodes on the consortium blockchain. The client verifies the legitimacy of the pseudo-root certificate based on its built-in CA root certificate. If it is valid, it can trust all the nodes trusted by the pseudo-root certificate, which are actually nodes in the consortium blockchain. It should be noted that the pseudo-root certificate is different to the system’s internal CA root certificate. The pseudo-root certificate is only used for communication in the DoK protocol and does not lead to the delegation of power.

The authoritative node will issue a set of temporary certificates to each node, each of which corresponds to a Certificate Identity (CID) and a corresponding asymmetric encryption algorithm. Therefore, knowing the CID means knowing which encryption algorithm the certificate uses to encrypt transmitted data. After receiving this set of temporary certificates, the client can select an encryption algorithm supported by both the server and client through these CIDs. Next, the client encrypts the data to be queried and its public key using the server’s public key, and the server decrypts it using its private key to obtain the data to be queried and the client’s public key. Finally, the response is encrypted with the client’s public key and sent back. The workflow of the DoK protocol is shown in Figure 7.

Before transmission begins, the client needs to obtain the server’s temporary certificate, which contains the server’s public key. This process does not occur during the initialization phase of establishing a connection, but rather after the client has configured the resolver. In Linux operating systems, this process occurs after the /etc/resolv.conf file has been modified. After obtaining and configuring the temporary certificate successfully, the client can communicate with the server via the DoK protocol. The entire communication process can be divided into three steps:Step 1The client sends the message to the server.1.1The client first checks whether the temporary certificate and the client’s own public and private key pairs are expired. If they are expired, the client requests a new temporary certificate from the server or generates its own public and private key pairs.1.2The client appends its public key to the message, which is the “Client Public Key” field in Figure 6a.1.3The client encrypts the entire message with the server public key in the temporary certificate.1.4The encrypted message is sent to the server.1.5The client waits for the ACK response from the server. If the ACK response is not received for a long time, the timeout retransmission mechanism is triggered and the client resends the message to the server.Step 2The server receives the message and responds to the client.2.1The server first checks whether a corresponding connection has been created based on the sender’s IP address, port number, and conv number. If not, a new connection is created.2.2The server responds with an ACK to indicate that it has received the message from the client.2.3The server decrypts the message with its own private key to obtain the record to be queried and the client’s public key.2.4The server queries the corresponding record.2.5The server encrypts the query result with the client’s public key.2.6The encrypted message is sent to the client.2.7The server waits for the ACK response from the client. If the ACK response is not received for a long time, the timeout retransmission mechanism is triggered and the server resends the message to the client.Step 3The client receives the response.3.1The client responds with an ACK to indicate that it has received the message from the server.3.2The client decrypts the message with its own private key to obtain the desired query result.3.3The client closes the connection.3.4When the server receives the ACK message from the client, it also closes the connection. The entire communication process is completed at this point.

### 4.3. Security and Performance Analysis of the DoK Protocol

In this section, we analyze the security and performance of the DoK protocol.

DNS flood attacks [2] are a type of DoS attack where the attacker sends a large number of query requests to the target DNS server, causing it to be unable to respond to legitimate DNS queries and resulting in the target system being unable to resolve domain names. DNS flood attacks can be avoided by combining the authentication mechanism in the DoK protocol and DNS-BC, as mentioned in Section 3.3. In addition, the DoK protocol itself has some resistance to DoS attacks, which is achieved through the VF field. As mentioned in Section 4.1, the VF field is used for source authentication, which can effectively detect zombie networks and blacklist the bots’ IP addresses.

DNS amplification attacks [3], DNS cache poisoning [4], DNS hijacking [5], and other attacks all exploit the vulnerabilities of the unencrypted and unauthenticated Do53 protocol. However, since the DoK protocol can authenticate the identities of both communicators using their public keys and the data are encrypted, these attacks can be effectively prevented.

Next, let us analyze the performance of the DoK protocol and see how it overcomes the performance bottleneck mentioned in Section 2.4.

As shown in Figure 7, the client first sends a message to the server, consuming a time of $0.5t_{RTT}$. After receiving the request, the server queries the corresponding record, consuming a time of $t_{query}$. The server then sends an ACK and the query result, consuming a time of $0.5t_{RTT}$. The client sends an ACK to the server after receiving the response, consuming a time of $0.5t_{RTT}$. Therefore, the total time consumed by the entire communication process is: (5)$$t_{DoK\_total}=1.5t_{RTT}+t_{query}$$

In fact, the client has already received the desired response before responding with an ACK to the server. Therefore, for the client, the time consumed from initiating the request to obtaining the desired data is: (6)$$t_{DoK}=t_{RTT}+t_{query}$$

It can be seen that the query time of the DoK protocol is significantly shorter than that of the DoT protocol described in Formula (Equation 1). In our experiments, the average query time of the DoK protocol is one quarter of that of the DoT protocol.

Another issue of concern to us is that since asymmetric encryption is more time consuming than symmetric encryption, will this cause performance bottlenecks for the DoK protocol?

In fact, the answer is no. Even though asymmetric encryption is much slower than symmetric encryption, it does not cause performance bottlenecks, as the total time for a DNS query is typically between 10 ms and 1000 ms [11], and the encryption and decryption time for asymmetric encryption is usually less than 1 ms [26]. In our experiments, the DoK protocol was able to significantly reduce query times even when using asymmetric encryption to encrypt transmitted data. The experimental results of this part will be shown in Section 5.3.

## 5. Experiments

### 5.1. Implementation

We have implemented a prototype of DNS-BC using Golang 1.20.6 and Python 3.10.12. The prototype is built on Hyperledger Fabric v2.4.9, with smart contracts written in Golang. The data structure stored in the blockchain includes DNS cache records and the credibility score of nodes. We have implemented two architectures: Monolithic Architecture (MA) and Service-Oriented Architecture (SOA) [27]. In MA, all the data are maintained together, and each row of data for Create, Read, Update, and Delete (CRUD) operations includes the domain name, record, owner, and owner’s credibility score. In SOA, record management and credibility score management are separated, providing two services: a record management service and a credibility management service. The CRUD operations of the record management service handle the domain name, records, and owner data, while the CRUD operations of the credibility management service handle the owner and the owner’s credibility score data.

To compare with traditional DNS cache systems, we referred to the implementations of DNS caches in two commonly used DNS services in Linux, bind [28] and dnsmasq [29], and simulated a traditional DNS cache system. This system stores cache records in a hash table created in computer memory, where the key of the hash table is the domain name and the value is the record. The CRUD operations of the hash table achieve O(1) time complexity. This part of the code is written in Python3.

We have implemented prototypes of four transport protocols for analysis and comparison using Python3: raw TCP without encryption, DoK, TCP + TLS, and HTTP/1.1 + TLS. The encryption part of the DoK protocol uses eciespy, a Python library that implements the secp256k1 elliptic curve used in Bitcoin’s public key cryptography. The certificates used in TLS were generated using OpenSSL.

The source code is completely open source and hosted on GitHub: https://github.com/sainnhe/DNS-BC (accessed on 6 July 2023).

### 5.2. Experimental Environment

In order to verify the feasibility and performance of DNS-BC and the DoK protocol, we conducted some experiments. The experimental environment was as follows:The services were deployed on Amazon Web Services (AWS) Lightsail. Unless otherwise specified, the server has the following technical specifications:–Operating system: Ubuntu 20.04.6 LTS (Focal Fossa)–RAM: 512 MB–Swap: 2 GB–CPU: 1 vCPU, x86_64–SSD: 20 GB–Location: Oregon, Zone A (us-west-2a)The client device was a MacBook Air. Unless otherwise specified, the client has the following technical specifications:–Operating system: macOS 13.3 (Ventura)–RAM: 16 GB–Swap: 1 GB–CPU: Apple M1, 8 cores, aarch64–SSD: 512 GB–Location: Shaanxi, China–Network: Wifi, China Unicom

### 5.3. Results and Analysis

One of our primary concerns is how the performance of DNS-BC compares to that of a traditional DNS caching system. To investigate this, we tested the probability distribution of the query time of both systems and the results are presented in Figure 8. It can be seen that the query time of DNS-BC is longer than that of the traditional DNS caching system because the former needs to retrieve query results over the network, while the latter can directly obtain results from the memory. Additionally, verifying the ledger in the blockchain when querying also incurs some performance loss. Overall, the performance of DNS-BC is slightly worse, but the difference is not significant for users.

We have implemented two architectures, namely MA and SOA. MA is a traditional software architecture pattern where all application components are packaged together in a single software package. The main characteristic of this architecture pattern is tight coupling, meaning that any change in one component may affect the entire application. However, due to the close relationships between components, it usually performs well. On the other hand, SOA is a service-based architecture pattern where the application is decomposed into a series of independent services. Each service is a standalone functional unit that can be developed, deployed, and scaled independently. In this architecture, the coupling between components is low, making it easier to maintain, and it has a stronger robustness. However, communication between components introduces an additional overhead, resulting in a lower performance compared to MA. Figure 9 compares the performance difference between the two architectures we have implemented, showing that MA performs slightly better than SOA, but the overall impact is not significant.

The system we designed is based on Hyperledger Fabric, which has strong scalability. We can easily add or remove nodes to the network. In Figure 10, we compare the performance impact of different numbers of blockchain nodes and different number of clients. It can be seen that changes in the network scale do not have a significant impact on the query latency. This is because we have built a distributed network, and as the scale expands, the traffic that needs to be processed by individual nodes does not increase sharply, so there is not a significant impact on performance.

In DNS-BC, the accuracy of the cache can reach 100% and a high real-time performance can be achieved by eliminating the non-real time caused by expired caches. This is achieved by increasing the cache hit rate, as we analyzed in Section 3.3. The experimental results in Figure 11 show that the higher the proportion of caches owned by name servers, the higher the cache hit rate, and when all caches are maintained by name servers, the cache hit rate is 100%. In this case, the accuracy of the cache can be considered as 100%, and the non-real time caused by expired caches is eliminated.

DNS-BC can protect valuable nodes, namely name servers, from DoS attacks by filtering traffic, and name servers only process requests from other consortium blockchain nodes. The more caches hit, the fewer requests the name server maintaining the cache needs to process, thereby effectively protecting these nodes. The experimental results in Figure 12 show that the higher the proportion of caches owned by name servers, the fewer requests name servers need to process. When all caches are maintained by name servers, the number of requests that name servers need to process is 0, and all requests are handled by recursive resolvers, thereby preventing DoS attacks on name servers. As analyzed in Section 3.3, since recursive resolvers are typically maintained by CDN providers or cloud service providers such as Cloudflare and Google, they have strong resistance to DoS attacks, so the additional processing of these requests does not have a significant impact on them.

To compare the performance of the DoK protocol with common encrypted DNS protocols, we simulated data transmission using the DoT and DoH protocols and compared the results, as shown in Table 1. The client was located in China and the server was located in the United States. It can be seen that the average query time of the DoK protocol is approximately a quarter of that of the DoT and DoH protocols, indicating that our newly designed encrypted DNS protocol performs much better in terms of the transmission delay than DoT and DoH.

The relative standard deviation represents the high or low packet loss rate, which is triggered by the protocol’s retransmission mechanism when packet loss occurs, resulting in query times that are far longer than the average time. Therefore, the more packet loss that occurs, the higher the relative standard deviation. It can be seen that the packet loss rate of DoT is higher than that of DoH, and the packet loss rate of DoK is much higher than both of them. This is because packets pass through many routing nodes and firewalls during transmission, and less common packets are more likely to be discarded. For example, when a packet passes through a firewall that only allows HTTPS traffic, the packets of the DoT and DoK protocols are discarded. Since the DoH protocol disguises packets as normal HTTPS traffic, the packet loss rate is low, while the DoT and DoK protocols are relatively less common, resulting in higher packet loss rates. Since the DoK protocol we have designed is a new protocol, it is easy to be filtered out by firewalls, leading to a much higher packet loss rate than DoT and DoH.

Due to the strict censorship of network traffic in China, we conducted another experiment to demonstrate that a smoother network environment can reduce packet loss rates. In this experiment, the client was located in the eastern United States and the server was located in the western United States, and the results are shown in Table 2. It can be seen that compared with Table 1, the relative standard deviation is significantly reduced, indicating a much lower packet loss rate, which is caused by the filtering of traffic by routing nodes. The smoother the network environment, the lower the packet loss rate.

Another issue we are concerned about is the impact of using asymmetric encryption on the transmission performance, as asymmetric encryption is typically more time consuming. To investigate this, we conducted an experiment to test the time taken to transmit different lengths of messages using the DoK protocol. The experimental results are shown in Figure 13. We tested the transmission time for messages of 8 bytes, 128 bytes, and 1024 bytes. Since DNS records are usually not very long, the range of 8 to 1024 bytes covers the majority of transmission requirements. It can be seen that the length of the message does not significantly affect the transmission time. Therefore, the DoK protocol can effectively handle DNS transmission requirements.

Furthermore, to compare the impact of different encryption methods on the transmission performance, we compared the effects of no encryption, symmetric encryption using AES256, and asymmetric encryption using ECC as adopted in the DoK protocol. The experimental results are shown in Figure 14, which indicates that there is no significant difference in the transmission performance among the three methods. This also confirms our analysis in Section 4.3 that the network transmission latency is several orders of magnitude higher than the time spent on encryption and decryption. Therefore, the performance bottleneck lies in network transmission rather than encryption and decryption. Even with the use of asymmetric encryption, the DoK protocol can still achieve a good performance.

### 5.4. Discussion of the Experiments

In the above experiments, we conducted functional and performance verifications of the DNS-BC caching system and the DoK protocol. The experimental results show that compared to traditional DNS caching systems, DNS-BC has slightly longer query times, but the difference is not significant for users. The system implemented based on MA has a better performance compared to the system implemented based on SOA, but it is more difficult to maintain. DNS-BC adopts a distributed architecture, which makes it highly scalable and capable of achieving a high accuracy and a real-time performance, while effectively protecting name servers from DoS attacks.

Our designed DoK protocol has a query speed four times that of DoT and DoH, but the packet loss rate is higher than both of them, which is caused by the network environment. Since the DoK protocol is a new protocol, its transmission traffic is very rare, so it is easy to be discarded by a routing node. However, we have proven that the DoK protocol can also achieve a very low packet loss rate in a smooth network environment. Although DoK uses asymmetric encryption, it still achieves a good performance, and the message length and encryption method have little impact on the transmission time.

## 6. Conclusions

In this paper, we propose a DNS caching system based on a consortium blockchain, which can achieve up to 100% accuracy, eliminate non-real time caused by expired caches, and protect valuable nodes in the system from DoS attacks. The experiments show that the more nodes maintained by name servers in the system, the less traffic the name servers need to handle. When all caches in the system are maintained by name servers, the accuracy of these caches can reach 100%, the non-real time caused by expired caches is eliminated, and name servers do not need to handle any traffic from recursive resolvers.

To further accelerate the data transmission speed, we designed a new encrypted DNS protocol, namely DoK, which breaks through the performance bottleneck of the widely used encrypted protocols DoT and DoH by reducing the number of handshakes during connection establishment. The experimental results show that the transmission speed of the DoK protocol is about four times that of the DoT and DoH protocols. The DoK protocol is designed to be compatible with existing DNS systems and can replace DoT and DoH in traditional DNS systems.

In future research, we will focus on improving the architecture of the caching system to adapt to large-scale distributed scenarios and comparing the impact of different consensus protocols on the performance. To further accelerate the transmission speed of the DoK protocol, we can compare different asymmetric encryption algorithms and select the most suitable ones to apply to the DoK protocol, so that message fragmentation can be minimized while ensuring security and a high encryption speed. 

## Figures and Tables

**Figure 1 sensors-23-06366-f001:**
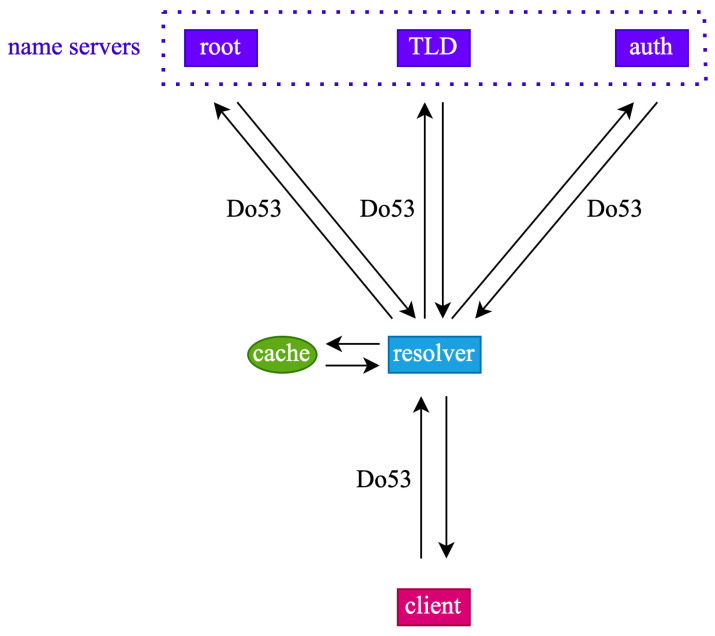
Typical architecture of a DNS system.

**Figure 2 sensors-23-06366-f002:**
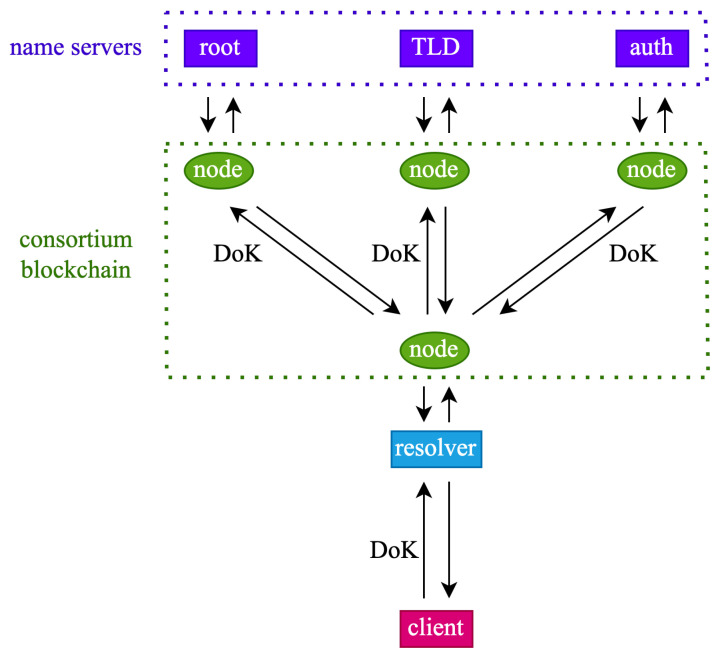
Architecture of DNS-BC.

**Figure 3 sensors-23-06366-f003:**
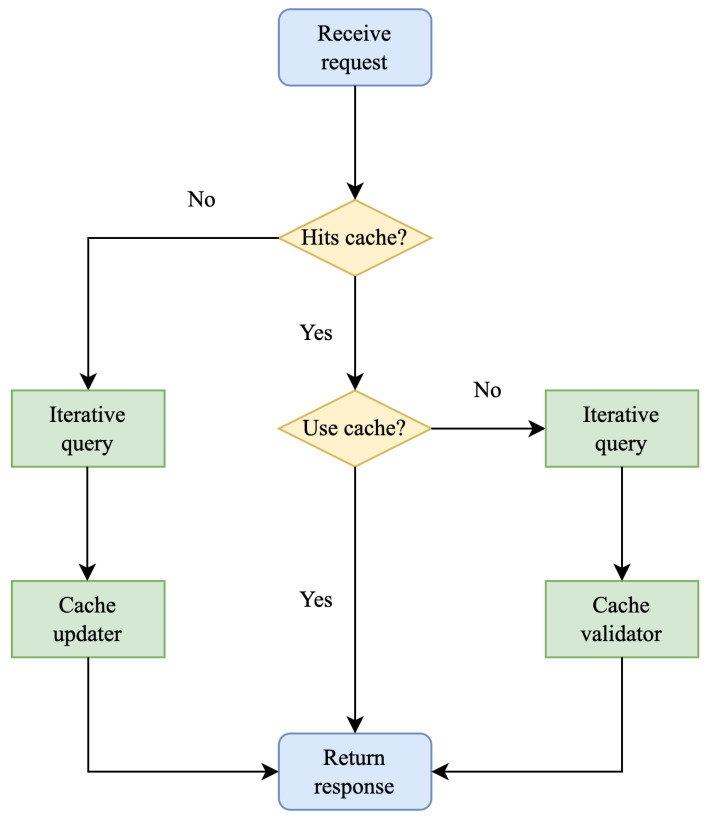
Workflow of the recursive resolvers in DNS-BC.

**Figure 4 sensors-23-06366-f004:**
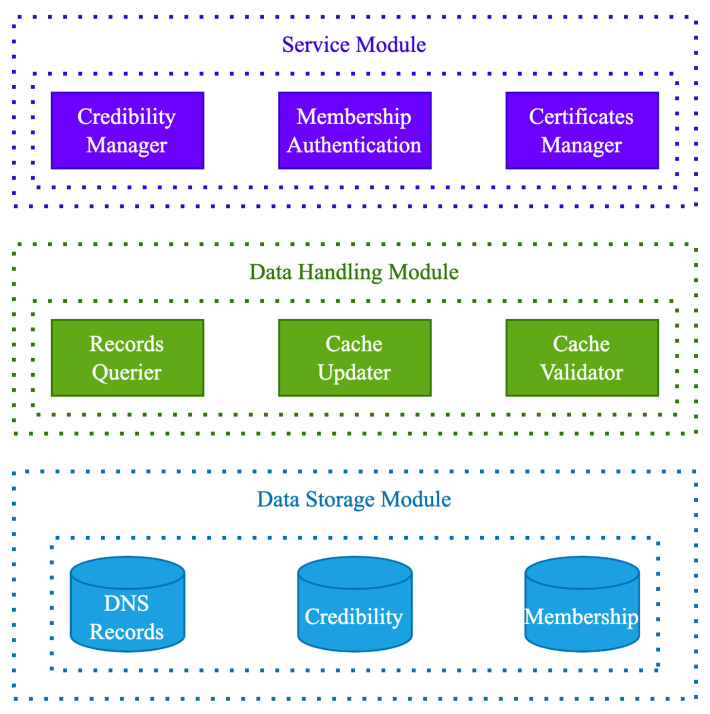
Function modules of the caching system.

**Figure 5 sensors-23-06366-f005:**
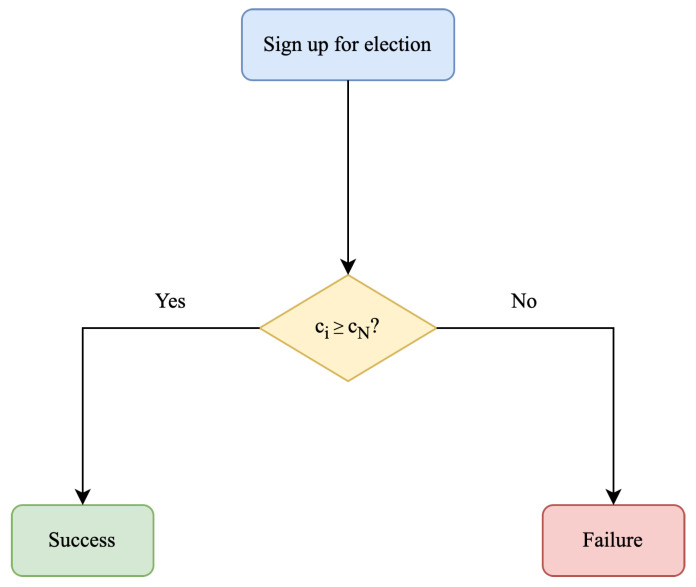
Electing authoritative nodes.

**Figure 6 sensors-23-06366-f006:**
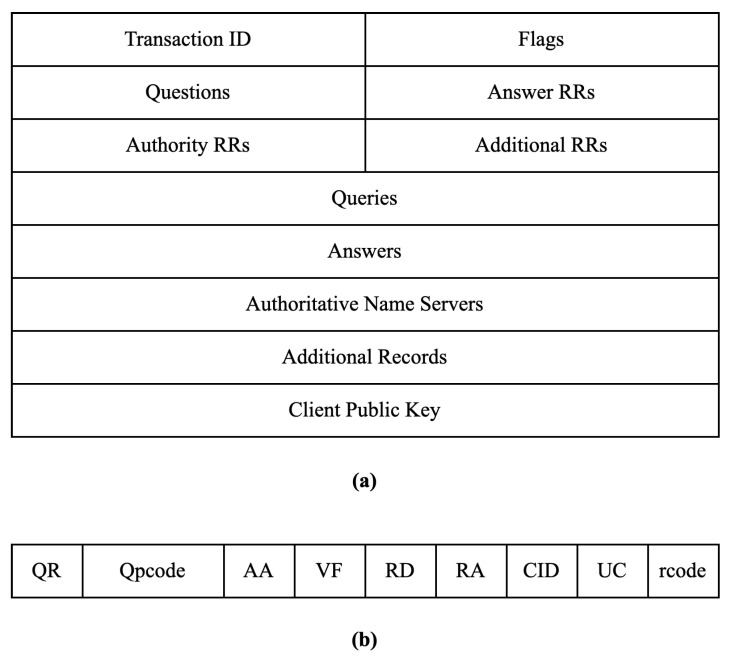
The format of the DoK message. (**a**) DoK message format; (**b**) flags field.

**Figure 7 sensors-23-06366-f007:**
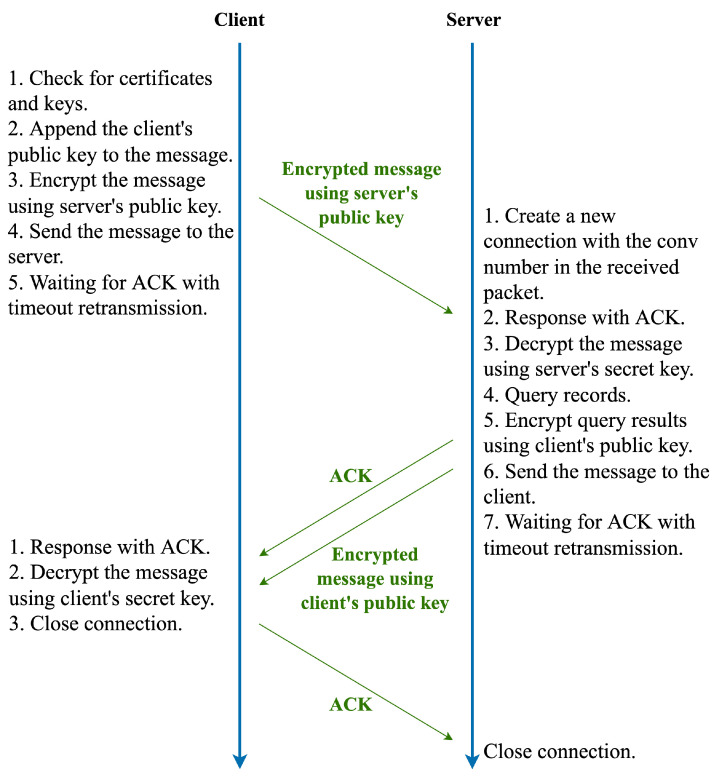
Workflow of the DoK protocol.

**Figure 8 sensors-23-06366-f008:**
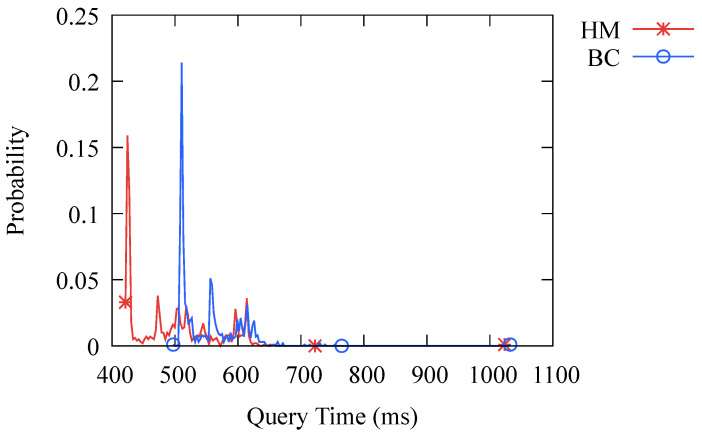
Probability distribution of query time. HM stands for Hashmap, which is the hashmap-based traditional caching system. BC stands for Blockchain, which is the consortium-blockchain-based caching system DNS-BC.

**Figure 9 sensors-23-06366-f009:**
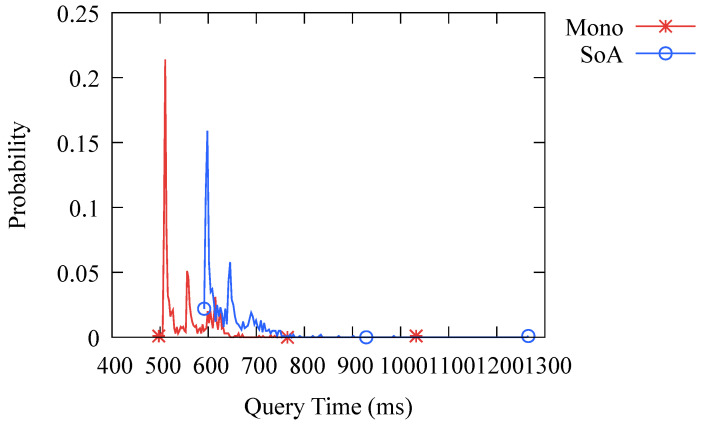
Probability distribution of query time in different architectures.

**Figure 10 sensors-23-06366-f010:**
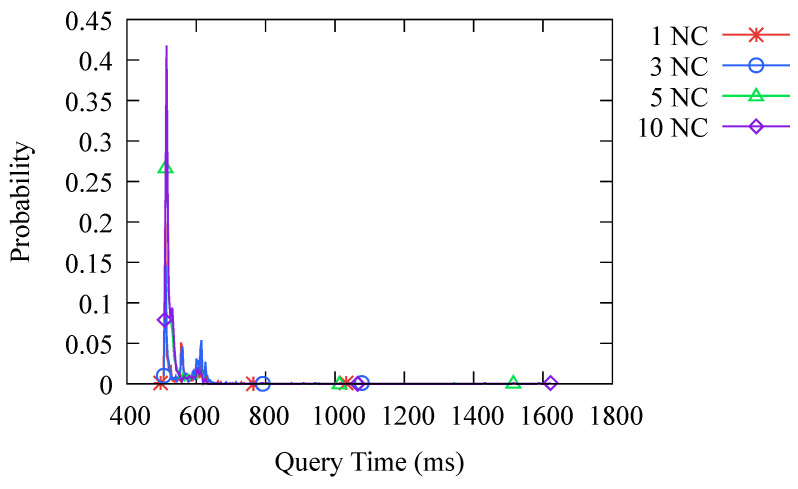
Probability distribution of query time at different network scales. NC stands for Node and Client, which can be used as a unit to describe the network scale.

**Figure 11 sensors-23-06366-f011:**
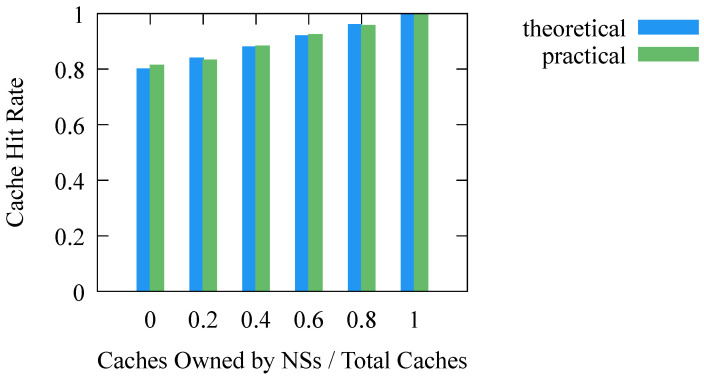
Cache hit rate. NSs stands for Name Servers.

**Figure 12 sensors-23-06366-f012:**
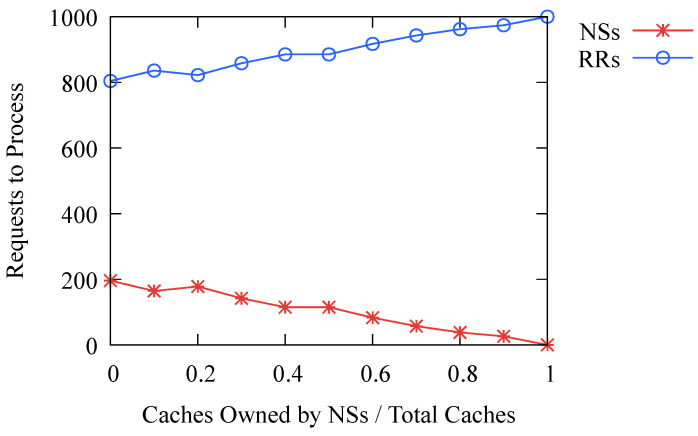
Number of requests to process in 1000 requests. NSs stands for Name Servers and RRs stands for Recursive Resolvers.

**Figure 13 sensors-23-06366-f013:**
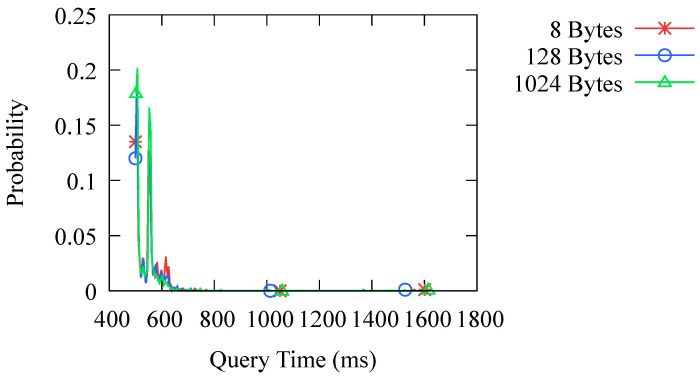
Probability distribution of query time with different message lengths.

**Figure 14 sensors-23-06366-f014:**
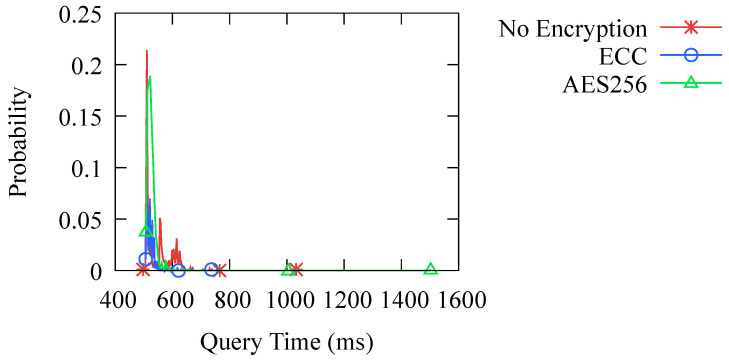
Probability distribution of query time with different encryption methods.

**Table 1 sensors-23-06366-t001:** Benchmark of different protocols. The client was located in Shaanxi, China, and the server was located in Oregon, US (AWS us-west-2). The system setup is described in Section 5.2.

Protocol	Average Query Time (ms)	Relative Standard Deviation ^1^
DoT	1045.522	21.916%
DoH	956.015	18.852%
DoK	249.545	35.784%

^1^ In probability theory and statistics, the coefficient of variation (CV), also known as the relative standard deviation (RSD), is a standardized measure of dispersion of a probability distribution or frequency distribution. It is defined as the ratio of the standard deviation σ to the mean μ. $RSD=\frac{\sigma }{\mu }$.

**Table 2 sensors-23-06366-t002:** Benchmark of different protocols. The client was located in Ohio, US (AWS us-east-2), and the server was located in Oregon, US (AWS us-west-2). The system setup is described in Section 5.2.

Protocol	Average Query Time (ms)	Relative Standard Deviation ^1^
DoT	205.569	1.886%
DoH	205.492	1.835%
DoK	53.560	3.085%

^1^ In probability theory and statistics, the coefficient of variation (CV), also known as the relative standard deviation (RSD), is a standardized measure of dispersion of a probability distribution or frequency distribution. It is defined as the ratio of the standard deviation σ to the mean μ. $RSD=\frac{\sigma }{\mu }$.

## Data Availability

Publicly available datasets were analyzed in this study. These data can be found here: https://github.com/sainnhe/DNS-BC (accessed on 6 July 2023).

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
