# Peer review of "DNS-BC: Fast, Reliable and Secure Domain Name System Caching System Based on a Consortium Blockchain"

_sensors, 2023, doi:10.3390/s23146366_

Round 1

Reviewer 1 Report

The paper introduces a caching system called DNS-BC, which is based on a consortium blockchain. DNS-BC leverages the synchronization mechanism of the consortium blockchain to achieve high real-time performance. It utilizes the immutable nature of the consortium blockchain and incorporates a credibility management system designed by the authors to achieve up to 100% accuracy. my concerns are as follows:

How can you justify that your proposed solution achieves 100 percent accuracy, as they did not provide a piece of evidence regarding this figure in the results section? Justify it.

The term DoK is defined in the abstract, what is KCP. The authors need to revise the abstract by including some numerical results.

How to compute the credibility score?

Add the system setup in tabular form

 A flow chart is required to explain the election process for a node to become an authoritative node.

n/a

Reviewer 2 Report

The paper "DNS-BC: Fast, Reliable and Secure DNS Caching System Based on Consortium Blockchain" proposes a new approach to DNS caching using consortium blockchain. While the paper presents an interesting idea, some areas require further development.

Firstly, the authors should include a more comprehensive review of related literature. Specifically, works such as https://doi.org/10.1145/3376044.3376057 and https://doi.org/10.1109/TNSE.2021.3068788  should be included in their discussion. This discussion will help provide a more thorough understanding of the current state-of-the-art in this field. The caching problem is one aspect of DNS systems, so full blockchain-based DNS approaches should be discussed. Related to the previous comment, the authors must compare their proposed solution with existing systems such as Ethereum Name Service (ENS). This analysis is important as ENS is currently actively used and is effective for DNS resolution.

Secondly, the paper needs more clarity in explaining certain technical aspects of the experimental evaluation. The authors should provide more detailed descriptions of the architecture and implementation of their system to make it easier for readers to understand and, most importantly, replicate. Implementing a cache system with MySQL seems naive, as better implementations could provide better performance. The authors should provide the reference from where this implementation was taken. It should be compared with a state-of-the-art implementation or an industry-standard implementation. This same concern applies to the implementation of encryption algorithms. 

Thirdly, the evaluation section could benefit from more rigorous testing and analysis, particularly regarding scalability. The authors should consider conducting experiments on larger-scale networks or demonstrate the scalability and performance of their system. Furthermore, this evaluation should include SoA blockchain implementations, such as the one presented in https://doi.org/10.1145/3453142.3491288. Lastly, the evaluation of the proposed encryption methods needs more evaluation in terms of security and also, of overhead. Since the authors rely on the blockchain protocol for their proposal, what does this protocol introduce the overhead compared to a non-blockchain-based protocol? How does selecting a different consensus of the blockchain network impact the performance?

Round 2

Reviewer 2 Report

The authors have successfully addressed all my previous comments and concerns.   As a minor text editing, I'll suggest the authors to rename section 5.4 to something like "Discussion of the experiments " and section 6.0 to  "conclusions".
With this minor comment,  the current version of the manuscript would be ready to be accepted for publication.
